# The Incidence of Bracing Treatment on Static and Dynamic Baropodometric Parameters in Adolescent Idiopathic Scoliosis

**DOI:** 10.3390/children9111608

**Published:** 2022-10-22

**Authors:** Vito Pavone, Alessia Caldaci, Giulia Rita Agata Mangano, Fabrizio Di Maria, Flora Maria Chiara Panvini, Marco Sapienza, Andrea Vescio, Federico Roggio, Giuseppe Musumeci, Gianluca Testa

**Affiliations:** 1Department of General Surgery and Medical Surgical Specialties, Section of Orthopaedics and Traumatology, University Hospital Policlinico “Rodolico-San Marco”, University of Catania, 95123 Catania, Italy; 2Department of Biomedical and Biotechnological Sciences, Section of Pharmacology, University of Catania, 95123 Catania, Italy; 3Department of Biomedical and Biotechnological Sciences Anatomy, Histology and Movement Sciences Section, School of Medicine, University of Catania, 95123 Catania, Italy

**Keywords:** adolescent idiopathic scoliosis, Sforzesco brace, baropodometric analysis, postural balance, conservative treatment

## Abstract

Postural balance is an important but not well-studied concept in the treatment of adolescent scoliosis. The aim of this study was to assess whether conservative treatment with Sforzesco bracing induced negative perturbations on postural stability, as related to static, postural, and dynamic baropodometric indicators. Twelve subjects (12 females, aged 11–16) with moderate AIS, were selected among a group of 97 patients. Inclusion criteria were: (1) confirmed diagnosis of moderate AIS (Cobb angle of 21° to 35° for the primary curve); (2) thoracic or thoracolumbar primary curve; (3) skeletal immaturity with growth cartilage visible on pretreatment radiographs (Risser < 5); (4) chronological age between 11 and 16 years; and (5) Sforzesco bracing treatment. All patients underwent a physical examination and radiological measurements with anteroposterior and lateral scans. Static, postural, and dynamic assessments were performed twice by barefoot patients, with and without Sforzesco bracing. Comparison between demographic, anthropometric, and clinical data highlighted a homogeneity of the sample. We evaluated the point of maximum pressure with and without bracing and found no statistically significant differences (*p* value = 0.22). In postural measurements, the laterolateral oscillations, anteroposterior oscillations, and average speed of oscillations were evaluated, comparing measurements with and without bracing. There were no statistically significant differences, except for the mean rate of oscillation, which was slightly increased in the recordings with a brace compared to those without a brace, *p* value = 0.045. Our findings show no statistically significant differences (*p* > 0.05) in static, postural, and dynamic baropodometric indicators.

## 1. Introduction

Adolescent idiopathic scoliosis is a three-dimensional deformity, defined as a lateral deviation and axial rotation of the spine [1]. Body asymmetries in idiopathic scoliosis involve the trunk, pelvis, and lower limbs [2,3]. Moderate curves require bracing as the standard treatment method during skeletal growth to restore spinal misalignment, to maintain spinal balance [4], and to prevent progression of the deformity [5]. Different types of braces and treatment protocols for scoliosis have been used [6].

It is known that brace treatment may affect lower extremity biomechanics during functional activities such as standing and walking, caused by the restrictive nature of bracing with continuous pressure on the trunk for a long period of time, along with mobility restriction [7]. Postural balance is the ability to keep the body in equilibrium and gain balance after the shift of body segments [8]. The foot plays a critical role in maintaining biomechanical function of the lower extremities, which includes balance arrangement and stabilization during human locomotion [9,10,11,12]. Dynamic postural stability, an individual’s ability to maintain balance while transitioning from a dynamic to a static state, is important [13]. Both static postural stability and dynamic postural stability are a result of complex coordination of central processing from visual, vestibular, and somatosensory pathways, as well as the resultant efferent response [14]. It is also known that compensation involves dynamic phenomena, which are poorly-served by highly-coordinated patterns of muscle activation/deactivation, disseminated throughout the whole body, and called “postural adjustments” [15,16,17]. It was previously reported that long-term (6 months) spinal bracing generated changes in gait biomechanics with increased pelvis and hip motion, decreased stance phase time and cadence, and increased step length [7]. Other studies demonstrate decreased pelvis and hip mobility immediately [18] as well as one-year after bracing [19]. The effects of a spinal brace on foot biomechanics in relation to the locomotor mechanism have not been studied at all [20].

The aim of this study was to assess whether conservative treatment with the Sforzesco brace could cause negative perturbations on postural stability in relation to static, postural, and dynamic baropodometric indicators.

## 2. Materials and Methods

From November 2020 to October 2021, 12 subjects (12 females, aged 11–16) with moderate AIS, were selected among a group of 97 patients, consecutively recorded in the Section of Orthopedics and Traumatology of the University Hospital Policlinico San Marco, Catania, Italy.

The study was conducted in compliance with the principles of the Declaration of Helsinki. All patient guardians signed informed consent papers before inclusion in the study. The recruitment procedures allowed for twelve subjects (12 females, aged 11–16 years) with moderate AIS among a group of a total of 97 patients.

Selection criteria were: (1) confirmed diagnosis of moderate AIS (Cobb angle of 21° to 35° for the primary curve, according to the SOSORT classification [21]); (2) thoracic or thoracolumbar primary curve; (3) skeletal immaturity with growth cartilage visible on pretreatment radiographs (Risser < 5); (4) chronological age between 11 and 16 years; and 5) Sforzesco bracing treatment. Exclusion criteria were: (1) scoliosis due to known causes or other disorders/spine anomalies; (2) neurological and neuromuscular disorders. All patients, after clinical evaluation using the bend-forward test [22] and scoliometer measurement [23], underwent radiological assessment with anteroposterior and laterally erect radiography scans; scoliosis severity was assessed by measuring the Cobb angle [24] according to stereotactic radiosurgery (SRS) guidelines [21]. Patients’ vital characteristics were also recorded. Demographic and clinical data were included: gender, age, standing height, weight, BMI, menarche in female patients, the Risser score [25], brace model, and treatment duration (Table 1).

Indications for brace treatment were Cobb angles of 21° to 35°; the Sforzesco brace was used in all patients for 14 h per day [26], which was set at ≥ 20° during accelerated growth for 11- to 13-year-old patients [27]. Physical therapy and sports activities were suggested for all subjects [28].

The static, postural, and dynamic assessments were performed twice by barefoot patients, with and without the Sforzesco brace. Patients were also asked to refrain from wearing their braces 24 h before the study day to avoid carry-over effects of brace treatment on study measurements [29].

Data were collected using a baropodometric platform (T-Plate, Molinari), to perform a static, postural, and dynamic analysis. Subjects were asked to stand with their feet wide apart at 20°, arms at their sides, and with eyes open for static and postural assessment [30].

For static evaluation, the point of maximum pressure was assessed. For dynamic evaluation of gait, the center of pressure excursion index (CPEI) [31] was detected for the right and left foot. The CPEI for both feet is evaluated for gait assessment: it represents the distance calculated from the line that joins the start and end points of the pressure center and the center of pressure (COP) point in the forefoot, and 1/3 of the total length of the foot. For a normalized value, this term is subsequently divided by width of the foot at that point. CPEI = BC/AD. Multiplying this value by 100 allows us to establish the percent of CPEI. Postural assessment with a baropodometric test was performed with the patient barefoot and in the standard reference position, with an acquisition time of 10 s plus both feet simultaneously on the sensorized mat.

The following parameters were extracted by the stabilometric assessment: point of maximum pressure (P.max g/cm^2^), sway variations along the anteroposterior (Var. Ant mm) and laterolateral (Var. Lat mm) directions, and mean sway velocity (mm/s) and CPEI of both feet [32,33].

To test the possibility of a significant alteration wearing the brace, non-parametric tailed paired *t*-tests (Wilcoxon test) were performed. Pearson’s correlation coefficient r was computed to assess the links between static, postural, dynamic, and clinical parameters [34]. All statistical analyses were performed with the computing package GraphPad Prism Version 5.0 (GraphPad Software Inc., San Diego, CA, USA). Continuous data were presented as mean and standard deviation. The selected threshold for statistical significance was *p* < 0.05.

## 3. Results

Recruitment procedures allowed our research group to involve 12 patients in the study: 12 females with AIS diagnosis. The mean age was 13.4 ± 1.55 years (range = 11–16). mean height, body weight and BMI were 152.4 ± 14.42 cm (range = 125–170), 49.75 ± 7.01 kg (range = 35–56), and 21.6 ± 3.59 (range = 16.5–31.9). Patients were treated for 10.6 ± 6.72 months, mean Cobb angle was 26.08 ± 3.8, and the mean Risser value was 2.83 ± 1.06.

Comparison between demographic, anthropometric, and clinical data deemed relevant for the homogeneity of the sample highlighted the absence of statistically significant differences between patients.

The point of maximum pressure was evaluated in static measurements; 620 ± 65 g/cm^2^ (range = 515–726) in recordings without a brace and 640 ± 95 g/cm^2^ (range = 540–779) in recordings with a brace; there were no statistically significant differences (*p* value = 0.22) (Figure 1).

Postural measurements were evaluated for laterolateral oscillations, anteroposterior oscillations, and the average speed of oscillation; these were 2.6 ± 5.8 mm (range = 0.5–21) (Figure 2), 3 ± 5.4 mm (range = 0.5–19.7) (Figure 3) e 2 ± 0.97 mm (range = 0.9–3.4) (Figure 4) in the recordings without a brace and 2.9 ± 4.9 mm (range = 0.4–18) (Figure 2), 4.2 ± 4.9 mm (range = 0.6–17.6) (Figure 3) e 2.6 ± 1.6 mm (range = 0.9–5.7) (Figure 4) in recordings with a brace. There were no statistically significant differences except for the mean rate of oscillation, which was slightly increased in the recordings with a brace compared to those without a brace, with *p* value = 0.045.

In records without a brace, the left CPEI was 20 ± 14 (range = 1.1–42.7), the right CPEI 17 ± 8.9 (range = 5.8–34.2) while in the brace, left CPEI 14 ± 10 (range = 2.3–36.2), right CPEI 18 ± 8.5 (range = 5.5–33.4). There were no statistically significant differences in CPEI with/without a brace (*p* > 0.05). (Figure 5 and Figure 6).

All results explained above are summarize in Table 2.

All parameters were related to the BMI and Cobb angle, using the Pearson test, with none found to be statistically significant.

The following chart is an example of the correlation between the average speed in the brace and the BMI: r = 0.3086; *p*-value = 0.1645 (Figure 7).

## 4. Discussion

Postural balance is an important but not clearly studied concept for treatment of adolescent scoliosis. In this study, we evaluated effects of the Sforzesco brace on postural balance in a sample of adolescents with idiopathic scoliosis, comparing results with and without the brace to assess whether conservative brace treatment could lead to adverse disturbances in postural stability. Comparison between the parameters recorded with and without the brace showed there were no statistically significant differences (*p* > 0.05) in static, postural, and dynamic baropodometric indicators. It is known that adolescent patients with idiopathic scoliosis present postural instability [35] and a significant increase in plantar pressure compared to healthy subjects, accord to Lee J-U et al. [36].

Based on the analysis of all our data, the Sforzesco brace did not affect the point of maximum pressure, overlapping with recordings performed without the brace. In static conditions, normal feet distribute the weight more in the heel, where there is a maximum pressure peak about 2.6 times higher than the peak located under the second and third metatarsal [37].

While walking, the maximum peak of pressure is under the second metatarsal head, followed by the third metatarsal head and hallux. This variation in the distribution of maximum peaks is due to an increase of pressure along the forefoot during the step propulsion phase, necessary to promote advancement of the contralateral limb [38].

Laterolateral, anteroposterior postural oscillations and average speed were shown to be almost overlapping, compared with and without brace data, according to the purpose of our study. Several studies in the literature have shown that brace treatment can reduce oscillations in adolescent patients with idiopathic scoliosis. The absence of statistically significant differences is probably due to the complex multifactorial nature of idiopathic scoliosis.

Excursion of the center of pressure as well as the force curve tend to move laterally in a cavus foot, while they move more medially in a flatfoot. This shift causes a variation of the CPEI, which can take on different values within a very wide range, reaching values of about 0.30 in cases of severe cavus foot and negative values in the most severe forms of flatness. CPEI value was similar in the recording with and without bracing. Our patients have shown a CPEI value that fits the normal range: 6.2–19.2.

In a recent study, some authors pointed out that CPEI of patients with moderate and severe AIS was significantly higher than healthy controls [39]. Therefore, these patients must compensate the postural asymmetry caused by changes in the shape of the spine through the vestibular and somatosensory system, such as the proprioceptive system of the ankle and increased energy consumption, which helps maintain a stable posture while walking [14,40]. When patients with AIS are in a lying position, their balance is very similar to that of healthy people; while walking they often show some abnormalities, such as increased body swing.

Using the brace aims to preserve body structures, but it could also improve postural balance, including positive effects on psychological aspects. New non-invasive and radiation-free technologies could help physicians in the decision-making process to find the therapy that fits best. For instance, gait analysis can provide information about kinematic movements of the trunk, upper and lower limbs [41], as rasterstereography can monitor treatment improvements in a short time without harmful effects [42], while infrared thermography can yield new insights about the muscles of the convex and concave sides [43].

The brace is a safe tool for conservative treatment of adolescent scoliosis, which does not modify static or dynamic postural parameters. It is advisable to integrate treatment with adequate physiotherapy, preferably with a 1:1 ratio with the therapist as well as psychological support. There are some limitations in our study, due to the small number of participants and the lack of a healthy control group. Our results show the change in regional plantar pressure distribution and how this is not affected using Sforzesco brace. However, to design an intervention for AIS patients, more multidimensional and scientific investigations conducted with different types of bracing are required.

## 5. Conclusions

Postural balance is an important but not clearly studied concept in the treatment of adolescent scoliosis. Our findings show no statistically significant differences in baropodometric indices for patients in treatment while using the Sforzesco brace.

## Figures and Tables

**Figure 1 children-09-01608-f001:**
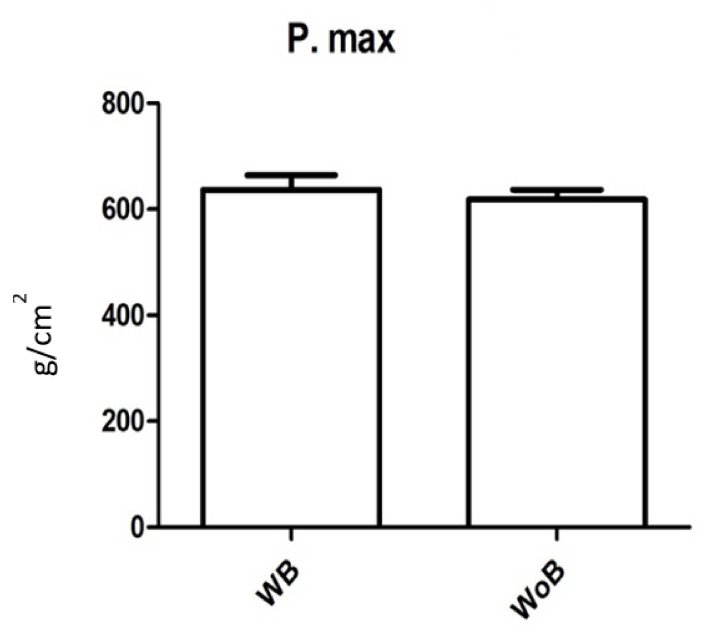
Evaluation of the point of maximum pressure (g/cm^2^) in static measurements with (WB) and without (WoB) brace. Error bars represent Standard Error (SE).

**Figure 2 children-09-01608-f002:**
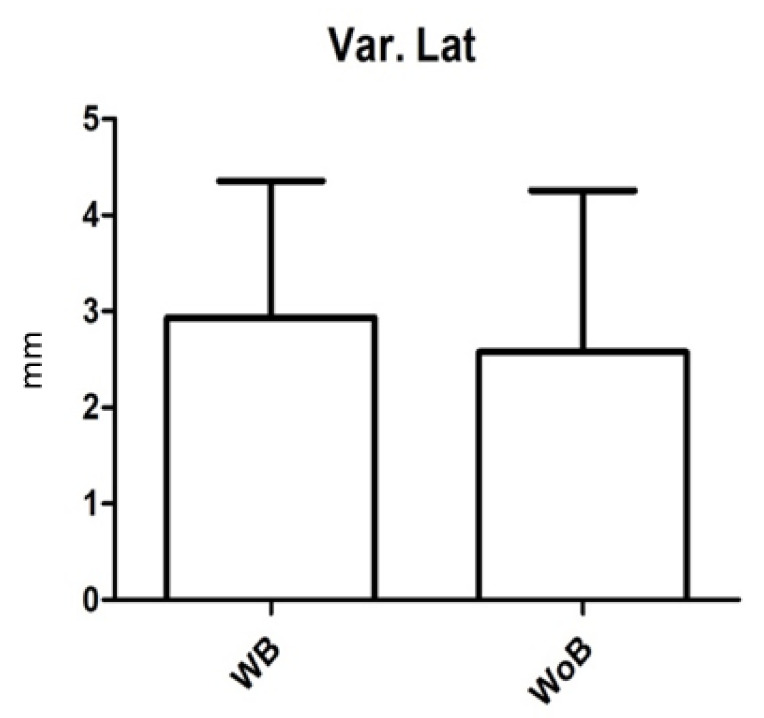
Variation of the laterolateral oscillation (mm) in the baropodometric measurement with (WB) and without (WoB) a brace. Error bars represent Standard Error (SE).

**Figure 3 children-09-01608-f003:**
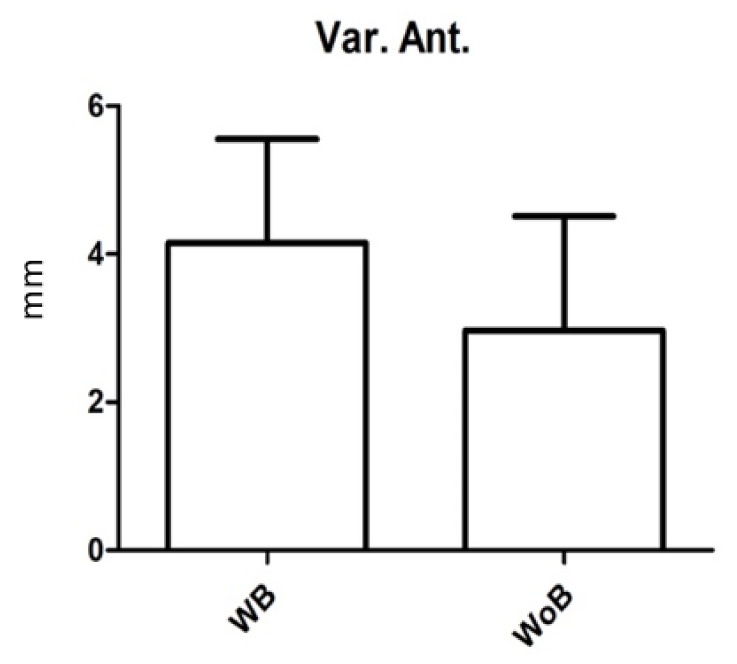
Variation of the anteroposterior oscillation (mm) in the baropodometric measurement with (WB) and without (WoB) a brace. Error bars represent Standard Error (SE).

**Figure 4 children-09-01608-f004:**
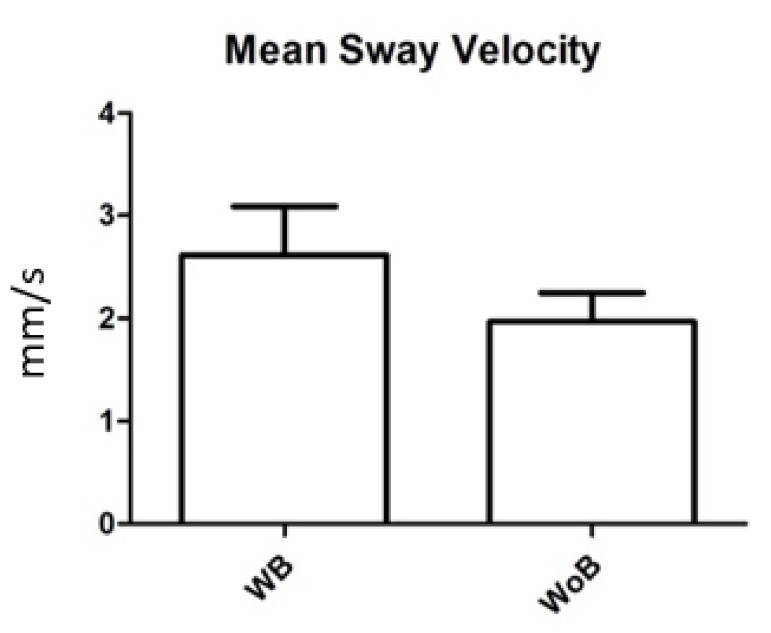
Mean sway velocity (mm/s) in the baropodometric measurement with (WB) and without (WoB) a brace. Error bars represent Standard Error (SE).

**Figure 5 children-09-01608-f005:**
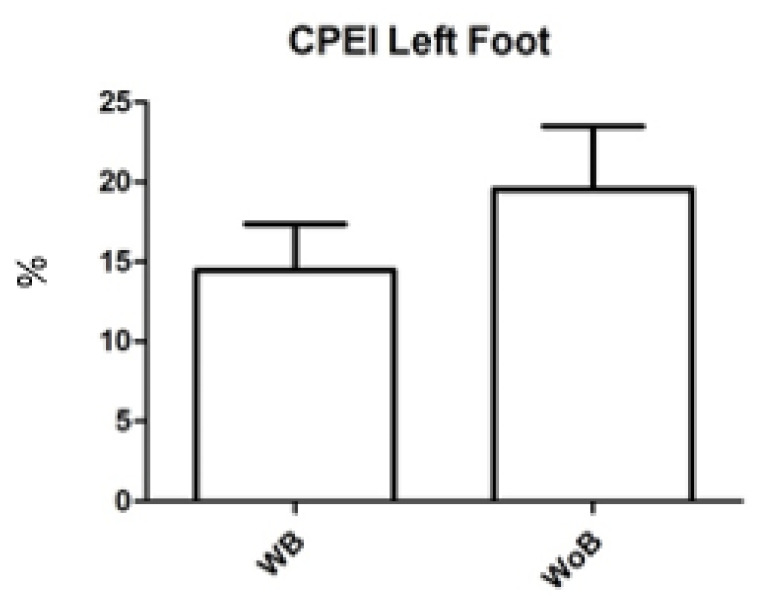
CPEI Left Foot with (WB) and without (WoB) brace.

**Figure 6 children-09-01608-f006:**
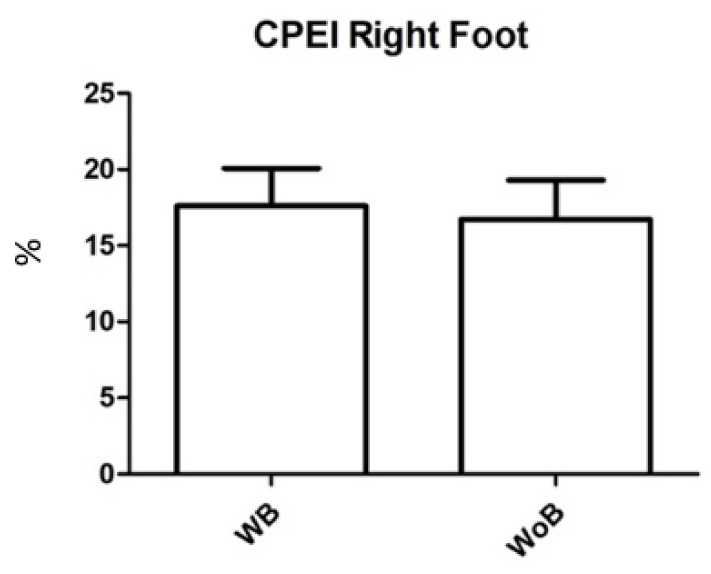
CPEI Right Foot with (WB) and without (WoB) brace.

**Figure 7 children-09-01608-f007:**
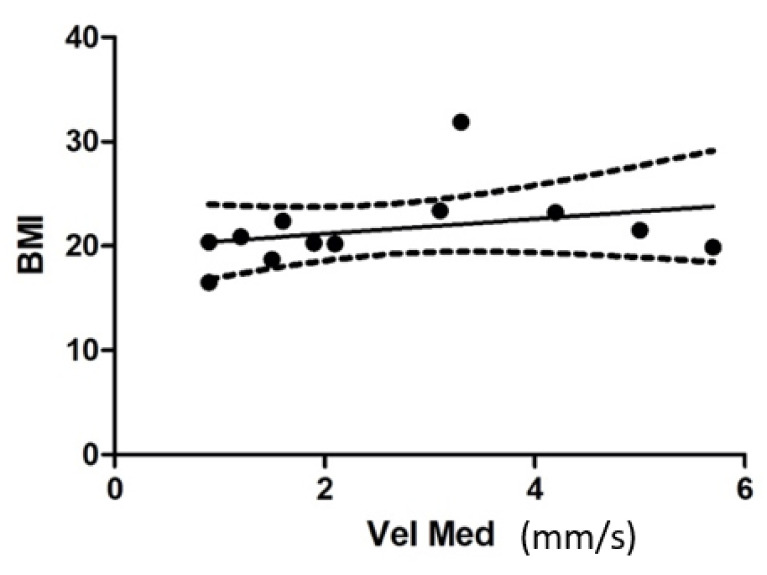
Correlation between the average speed (mm/s) with brace (WB) and the BMI. Dashed lines are 95% CIs.

**Table 1 children-09-01608-t001:** Means and standard deviations of measurements recorded in static, postural, and dynamic assessments with and without the brace as well as the relative *p* value. Point of maximum pressure (P. max); Var Lat. (lateral variation); Var. Ant (anterior variation); Vel. Mean (mean velocity); CPEI (center of the pressure excursion index).

	STATIC	POSTURAL			GAIT ANALYSIS	
	P. max	Var. Lat	Var. Ant.	Vel. Mean	CPEI SX	CPEI DX
NO BRACE	620 (±65)	2.6 (±5.8)	3 (±5.4)	2 (±0.97)	20 (±14)	17 (±8.9)
BRACE	640 (±95)	2.9 (±4.9)	4.2 (±4.9)	2.6 (±1.6)	14 (±10)	18 (±8.5)
WILCOXON TEST (*p* value)	0.2277	0.2852	0.0881	0.0458	0.2065	0.1902

**Table 2 children-09-01608-t002:** Patients’ vital characteristics: Demographic and clinical data.

	PATIENTS	AGE (YEAR)	GENDER	HEIGHT (CM)	WEIGHT (KG)	BMI	TIME OF TREATMENT (MONTHS)	COBB ANGLE (DEGREE)	RISSER SCORE
1	S. V.	13	F	165	55	20.2	12	28	3
2	S. G.	15	F	159	53	20.9	14	32	4
3	S. G.	16	F	170	54	18.7	24	24	4
4	G. P.	13	F	155	49	20.4	8	21	3
5	C. S.	13	F	152	55	23.4	9	22	3
6	B. S.	11	F	135	58	31.9	4	23	1
7	C. C.	13	F	152	46	19.9	7	26	2
8	C. M.	15	F	163	54	20.3	22	30	4
9	C. M.	11	F	125	35	22.4	2	21	1
10	P. C.	15	F	165	45	16.5	13	32	4
11	S. A.	12	F	128	38	23.2	2	28	2
12	U. A. T.	14	F	160	55	21.5	10	26	3
MEAN		13.4		152.4	49.75	21.6	10.6	26.08	2.83
SD		1.55		14.42	7.01	3.59	6.72	3.81	1.06
MIN.		11		125	35	16.5	2	21	1
MAX.		16		170	58	31.9	24	32	4

## Data Availability

Not applicable.

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
