# Peer review of "The Incidence of Bracing Treatment on Static and Dynamic Baropodometric Parameters in Adolescent Idiopathic Scoliosis"

_children, 2022, doi:10.3390/children9111608_

Round 1
Reviewer 1 Report
Dear Editor,
Thank you for the opportunity to review this paper.
Abstract - ok
Introduction - ok
Material and Methods
Only 12 subjects included, represents a small number, more of a case presentation than a representative cohort.
Statistics should also include a power study and required minimum number of patients in order to be statistically significant.
Results
Lines 139-154 are very precise and descriptive, no need for plots figures 1-6, especially due to the fact that the results have no statistically significance. I would recommend removing those plots.
Table 1 and Figure 7 are ok.
Discussions
Does phisiotherapy alter patient's static and dynamic baropodometric parameters? Have all the patients done the same type of phisiotherapy (eg. schrot exercises)? Was the experience worse for patients that had flat-feet, or all of them had normal feet? Standing AP and lateral X-ray of the foot?
Limitations of the study were presented.
What are the strenghts of the study?
What should further studies search for?
Please elaborate
Conclusions
What does the paper add new to the medical practice of an orthopedist that has scoliosis patients in care?
Author Response
Dear author,
Thank you for your time and for the requested revisions. We are glad you enjoyed our manuscript.
Material and Methods
Unfortunately, only 12 subjects are part of our cohort, we realize that it is not a very high number but we have collected parameters and values very precise.
Results
Physiotherapy was performed weekly with or without bracing by our patients, although it is not an evaluated baropodometric parameter. They all performed schrot’s method. Regarding foot evaluation, we evaluated the differences between both flatfoot and cavus foot: "Excursion of the center of pressure as well as the force curve tend to move laterally in a cavus foot, while they move more medially in a flatfoot."
Thank you for your comment, we would like to leave the figures to be clearer to the reader, while not having a statistical significance.
We have included what should be improved in future studies.
Conclusion
We believe that the baropodometric examination is important in the evaluation of the patient affected by AIS.
We hope to cooperate in the future.
Reviewer 2 Report
The authors studied postural balance in scoliosis patients and analysed the baropodometric parameters when the patients were wearing or not wearing the brace.
The results obtained from 12 subjects were mostly not significant in terms of statistical analysis. However, some interesting findings could be drawn from the data.
The report has been presented as a pre-print. It may be better to report this as an acknowledgment.
However, there are some areas to improve the presentation of the paper.
1. No unit is shown in all figures. For example, Fig 5, Y axis, the unit should be %.
2. Legends of figures. It should state what are the error bars representing, either the SD or SE. or something else.
3. Although the study did not have a control group (which was a major limitation by itself), some ideas about the expected normal values for each measurement could be put down into the figures or added into the discussion section.
4. Interpretation of CPEI may be enhanced. For example, it would be expected the "normal" values would be around 20%. So scoliosis patients were walking more naturally when they were not wearing the brace. (WoB). On the other hand, when they weared the Brace (WB), CPEI dropped in the left foot. (but not statistically significant).
5. more about point 4. An in-depth analysis of CEPI, such as analyzing L and R feet together, or similar ideas may lead to a better understanding of the postural adaptation to wearing the brace.
Author Response
Dear author,
Thank you for your time and for the requested revisions. We are glad you enjoyed our manuscript.
We have made changes to the figures as required in points 1 and 2.
In the discussion, we have included normal values for healthy patients, as requested in point 3.
In figures 5 and 6 you can find the difference of CPEI in the right and left foot.
We hope to cooperate in the future.